# The Impact of COVID-19 on the Oral Bacterial Flora in Patients Wearing Complete Dentures and on the Level of Exhaled Nitric Oxide as a Marker of Inflammation

**DOI:** 10.3390/jcm12175556

**Published:** 2023-08-26

**Authors:** Magdalena Wyszyńska, Aleksandra Czelakowska, Przemysław Rosak, Jacek Kasperski, Maria Łopacińska, Amir Ghanem, Anna Mertas, Małgorzata Skucha-Nowak

**Affiliations:** 1Department of Dental Materials, Division of Medical Sciences in Zabrze, Medical University of Silesia in Katowice, 15 Poniatowskiego Street, 40-055 Katowice, Poland; 2Department of Dental Prosthetics, Division of Medical Sciences in Zabrze, Medical University of Silesia in Katowice, 15 Poniatowskiego Street, 40-055 Katowice, Poland; aczelakowska@op.pl (A.C.); protstom@sum.edu.pl (J.K.); 3Specialist Dental Practice Przemysław Rosak, 13 Piłsudskiego Street, 41-300 Dąbrowa Górnicza, Poland; rosakowski@o2.pl; 4Institute of Dentistry and General Medicine, 8 Łabędzia Street, 40-534 Katowice, Poland; nowak.isia@gmail.com; 5Doctoral’s School, Division of Medical Sciences in Zabrze, Medical University of Silesia in Katowice, 15 Poniatowskiego Street, 40-055 Katowice, Poland; s75720@365.sum.edu.pl; 6Department of Microbiology and Immunology, Faculty of Medical Sciences in Zabrze, Medical University of Silesia in Katowice, 15 Poniatowskiego Street, 40-055 Katowice, Poland; amertas@sum.edu.pl; 7Department of Dental Propedeutics, Division of Medical Sciences in Zabrze, Medical University of Silesia in Katowice, 15 Poniatowskiego Street, 40-055 Katowice, Poland; mskucha-nowak@sum.edu.pl

**Keywords:** diagnosis, oral diseases, COVID-19, nitric oxide, inflammation process, inflammation markers, microbiome, oral cavity

## Abstract

Background: Exhaled nitric oxide is helpful in the diagnosis of the inflammation process. The study aimed to analyze the impact of the COVID-19 disease on the oral bacterial flora of patients using complete dentures with a diagnostic device that measures the level of NO in exhaled air. Materials and Methods: The study included patients using upper and lower acrylic complete dentures. All patients participating in the study were vaccinated against COVID-19. The patients were divided into two groups. A dental examination was conducted in each group. The NO concentration was measured using the Vivatmo Pro device. An oral microbiological examination was performed by taking a swab from the bottom of the mouth. Results: There were no statistically significant differences in the distribution of NO in relation to the number of bacteria from isolated families in the study and control groups and no statistically significant correlations between the level of NO and the number of bacteria from all families in the control and study group. Significantly higher NO values were present in the vaccinated and COVID-19-positive history population compared to the vaccinated and with no COVID-19 history population (patients with no clinical symptoms of infection or unaware they had COVID-19). Conclusions: There are statistically significant differences in NO distribution in the considered populations: vaccinated and sick, and vaccinated and with a negative history of COVID-19. The measurement of NO in exhaled air can be a complementary, non-invasive diagnostic and inflammation monitoring method.

## 1. Introduction

In the 90s of the last century, the concentration of nitric oxide (NO) in the exhaled air of animals and humans was first determined. In 1998, R.F. Furchgatt, L.I. Ignarro and F. Murad were awarded the Nobel Prize for “discovering the importance of NO as a messenger signal in the cardiovascular system”. The interest in measuring NO concentration in exhaled air as an indicator of inflammatory changes is growing. This is confirmed by reports on its participation in various functions of the respiratory system. The advantage of this test is its non-invasiveness and quick results [1,2].

NO is a molecule with a versatile effect. It participates in both pathological and physiological processes. The function of NO depends on its concentration, where it is produced and how it interacts with other molecules. Active nitrogen compounds have cytotoxic and immunoregulatory effects [3,4,5,6,7]. NO is produced in the inflammation process. Its formation is connected with induced nitric oxide synthase (iNOS), which stimulates the production of reactive oxygen species and pro-inflammatory prostaglandins. At the end of this pathway, blood vessels become dilated and permeable to albumin, which causes edema [8]. The measurement of the concentration of NO in the exhaled air is used, among others, in the diagnosis and monitoring of treatment of patients with inflammatory processes in the oral cavity, respiratory tract and digestive system. The measurement of NO concentration in the exhaled air is a non-invasive and easy diagnostic test, especially used in patients with asthma. NO in the exhaled air is a sensitive indicator of the inflammatory process, reacting quickly to the treatment or exacerbation of the disease. Elevated NO levels work very well with other markers of inflammation assessed in biopsy material, fluid obtained from bronchoalveolar lavage or induced sputum [9,10,11]. Pathological processes in the oral cavity, such as tooth decay, inflammation of periodontal tissues or inflammation of the mucous membrane, are a local reaction of the organism to pathogens. According to Sokołowska et al. [1], in the inflammatory process, blood cells and endothelial cells activated by antigens activate the “inflammation cascade”, leading to tissue damage [12,13,14].

The oral cavity is a dynamic, diverse, and unique environment for microorganisms [15,16]. It is assumed that there are over 750 species of bacteria in the human oral cavity, but it is believed that only about 50% of the bacteria found there are known [17,18], which is confirmed by the Human Oral Microbiome Database (HOMD) [19]. The diversity of the oral microbiome is affected by the use of prosthetic restorations, hygiene, temperature, pH, oxidation-reduction potential, salinity, and saliva, which provides nutrients and removes metabolic products and additionally contains enzymes, e.g., amylase, antibacterial peptides, and antibodies [20]. The analysis of the results of molecular tests of the salivary microbiome showed the presence of over 100 types of bacteria, most of them: *Streptococcus*, *Prevotella*, *Veilonella*, *Neisseria*, *Heamophilus*, *Rothia*, *Porphyromonas*, *Fusobacterium*, *Scardovia*, *Parascardovia* and *Alloscardovia* [21,22,23]. Loss of teeth and prosthetic restorations are factors influencing the change in the oral microflora. In the case of loss of all teeth and the use of extensive plate prostheses, new, favorable conditions for the development of microorganisms in the oral cavity are created. Loss of teeth reduces the number of oral bacteria compared to complete dentition. Qualitative changes may consist of a disturbance in the proportion of anaerobic and aerobic bacteria, with the predominance of the aerobic, which is related to better conditions for cleaning by saliva after tooth loss, a change in diet and the lack of recesses in periodontal pockets and cavities, which are a kind of natural place for accumulation of and microbial growth. The most favorable conditions for the increased growth of microorganisms are the mucosal denture plate and the mucous membrane of the hard palate and alveolar processes covered with the denture plate. Hygienic negligence of dentures and dentition improper handling of dentures lead to the retention of food residues, a breeding ground for bacteria and fungi, which are part of the denture plaque and dental plaque. With a low pH of saliva, the proliferation of bacteria in the oral cavity increases, which leads to chronic inflammation [24]. The material of the prosthesis–acrylic resin–with a microporous structure favors the penetration and accumulation of microorganisms. Acrylic biodegrades over time, is absorbent, and when it has a rough surface, it provides good retention for the denture plate [25]. The conditions in the oral cavity under the denture plate, especially the upper one (humidity, elevated temperature, lack of self-cleaning by saliva), also contribute to the multiplication of pathogens [26].

The outbreak of COVID-19 caused by SARS-CoV-2 (Severe Acute Respiratory Syndrome Coronavirus 2) prompts researchers to intensively search the diagnosis of the biology, pathogenesis, and transmission routes of this virus [27,28,29,30]. COVID-19 is transmitted by droplets, respiratory tract, direct contact, and possibly fecal-oral transmission [31,32,33,34,35]. COVID-19 may also cause long-term complications. Most common and still being explored are complications that may arise, including cardiovascular, neurological, psychological, hematological, pulmonary, dermatological, and other injuries [36]. Previous studies have shown several potential late complications possible for long-term COVID-19 infection; these include lung fibrosis, venous thromboembolism (VTE), arterial thromboses, cardiac thrombosis and inflammation, stroke, “brain fog”, dermatological complications, and overall mood dysfunctions [37]. 

The oral cavity, according to the latest scientific reports, may play a key role in the pathogenesis and course of SARS-CoV-2 infection. Among patients with SARS-CoV-2 infection, the most frequently observed symptoms in the oral cavity are taste disturbances in the mouth, impaired sense of smell, dry mouth, inflammation of the mucous membrane, burning sensation, and difficulty swallowing. Other pathological changes noted in patients with COVID-19 also included vesicular, macular, and erosive lesions of the oral mucosa. Accompanying symptoms are palate pain, tongue pain and numerous small ulcerations on the mucous membrane of the palate with concomitant pain. In addition, bullous lesions on the inner mucosa of the lip and exfoliative gingivitis are also observed [38,39,40,41,42,43,44,45]. The correlation between the concentration of NO in exhaled air and the oral microbiome during COVID-19 infection has not been studied so far.

## 2. Aim of the Study

The study aimed to analyse the complications of the COVID-19 disease on the oral bacterial flora of patients wearing complete dentures using a diagnostic device that determines the level of NO in exhaled air as an additional tool.

## 3. Materials and Methods

### 3.1. Research Material

The study included patients aged 46 to 74 using upper and lower acrylic complete dentures for no longer than five years and reported for control of previously used prosthetic restorations. There were 34 women and 16 men in the group of patients. All patients participating in the study were vaccinated against COVID-19 with generally available vaccines in Poland (Pfizer, Astra Zeneca, Moderna, Johnson&Johnsson). Participants were enrolled in the study by completing a general health questionnaire and underwent a standard dental examination. The study included healthy people who did not take any medications, did not smoke, did not have symptoms of stomatitis, and had proper denture hygiene. On the day of the examination, patients were at least 2 h after a meal, after standard home hygienization of dentures, without any symptoms from the respiratory system. The study was conducted in the morning. The patients were divided into two groups. The first group of 25 patients is the control group, which included people with a negative history of COVID-19 (patients with no clinical symptoms of infection or unaware they had COVID-19) and the second group, the study group, is 25 people who have recovered from COVID-19 (at least six months after illness with mild symptoms). The mucosa was examined and classified according to Newton’s stomatitis classification modified by Spiechowicz. The assessment criteria are as follows: 0—no changes in the mucosa, presence of subjective symptoms: burning, acute pain or dryness in the mouth, 1—localized inflammatory changes, most often in the area of the mouths of the palatine glands, 2—diffuse inflammation of the oral mucosa covered by prosthesis, sometimes inflammatory edema, 3—hypertrophic papillary mucositis, usually in the middle part of the hard palate. The hygiene status of the prosthesis was assessed according to the Budtz-Jørgensen Index for the plate of the upper complete dentures. Interpretation of results: 0—no plaque after blunt scraping, 1—plaque visible after blunt scraping, 2—moderate plaque accumulation visible to the naked eye, denture surface partially covered with visible coating, 3—abundant plaque, denture surface completely covered with visible plaque raid.

### 3.2. Measurement of NO Concentration in Exhaled Air

The NO concentration was measured using the Vivatmo Pro device (Bosch, Waiblingen, Germany). Measurement procedures are quick and easy, and test results are available immediately. The device has a NO filter that eliminates NO present in the atmosphere. The flow controller maintains a constant amount of exhaled air. Every measurement is checked, and an automatic test checks the level of the contour lines. The patient receives a disposable mouthpiece for examination [46]. The system is maintenance-free, which means there are no complicated calibration procedures. The data can be managed directly from the touch screen. It is a portable medical device for measuring the fraction of nitric oxide in exhaled air (FeNO), giving the results in parts per billion [47].

### 3.3. Microbiological Examination

Oral microbiological examination was performed by taking a swab from the bottom of the mouth, using sterile swabs, in tubes with AMIES transport medium, with charcoal (DELTALAB, Rubi, Spain). The following media from Biomerieux (Marcy l’Etoile, France) were used to cultivate microorganisms: Columbia agar, Schaedler K3, ChromID Candida and Genbag anaer kits for the cultivation of anaerobic bacteria. Species identification of the cultured microorganisms was carried out using the following sets of reagents: ENTEROtest 24N, NEFERMtest 24N, STREPTOtest 24, STAPHYtest, ANAEROtest 23, OXItest, PYRAtest and the computer program TNW lite 6.5 for the identification of microorganisms by Erba-Lachema (Brno, Czech Republic). Biochemical tests by Biomerieux (Mercy l’ Etoile, France) were also used: Katalase, Slidex Staph Kit, and API Candida. 

### 3.4. Statistical Analysis

The data used in the statistical analysis come from two groups of twenty-five people. The first group consists of people who are vaccinated and sick (the study group), and the second group consists of people who are vaccinated and not sick (the control group). Each patient was examined for bacterial flora in the oral cavity. Four families of bacteria were considered: G + Cocci, G − Cocci, G + Rods, and Gram − Rods, and the appropriate groups of bacteria were distinguished in each family. In each patient, the number of groups from which bacteria living in the oral cavity originate was determined. The results are presented in the tables, where: “0”—no groups of bacteria, “1”—bacteria from one group, “2”—bacteria from two groups, and “3”—bacteria from three groups within one family. 

Non-parametric tests based on ranks were used in the statistical analysis. Ranks were successive natural numbers assigned to the previously sorted in ascending order sample values. To compare the distribution of features, the Mann-Whitney U test was used in two independent populations, and the Kruskal-Wallis test was used to compare the distribution in three independent populations. In addition, the Spearman’s rank correlation coefficient and its significance test were selected to study the correlation between two ordinal features. The last test used in the analysis is the chi-square test of independence. This test is used to examine the relationship between two features with a finite number of variants (however, not less than 2). The results for the above-mentioned tests are in the respective tables, where the important column is the *p*-value column. Since all tests were performed at a significance level of 0.05, the *p*-value will allow the appropriate decision to be made as follows. If the obtained *p*-value does not exceed the significance level, i.e., the number 0.05, we reject the null hypothesis at the assumed significance level. Otherwise, we have no grounds to reject it at the assumed materiality level. The results of the analysis were divided into three parts: results concerning the study group, control group results, and results comparing the two groups. The statistical analysis was made with Statistica software, ver. 13.3. 

## 4. Results

### 4.1. Study Group—Patients Vaccinated with Positive COVID-19 History

#### 4.1.1. Distribution of NO in Relation to the Number of Bacteria from Isolated Families

Based on the Kruskal-Wallis test, the NO value was determined in each group, considering the division into the number of Gram + (cocci), Gram − (Cocci), Gram + rods (mycobacteria) and Gram − rods (mycobacteria) families. As shown in Table 1, there are no statistically significant differences in the distribution of NO in relation to the number of bacteria from isolated families in the population of vaccinated and ill people.

#### 4.1.2. Correlation between NO and the Number of Bacteria from Particular Families

The study of the correlation between the NO value and the recorded number of bacteria from the Gram + (cocci), Gram − (Cocci), Gram + rods (mycobacteria), Gram − rods (mycobacteria) families in the study group was performed using Spearman’s rank correlation coefficient. The significance test showed no statistically significant correlations between the level of NO and the number of bacteria from individual families in the population of vaccinated and ill people (Table 2).

### 4.2. Control Group—Patients Vaccinated, with Negative COVID-19 History

#### 4.2.1. Distribution of NO in Relation to the Number of Bacteria from Particular Families

Based on the Kruskal-Wallis test, the NO value was determined in particular groups, taking into account the division into the number of recorded bacteria from the Gram + (cocci), Gram − (Cocci) and Gram + rods (mycobacteria) families. There are no statistically significant differences in the distribution of NO with respect to the number of bacteria from isolated families in the population of vaccinated and non-vaccinated persons (Table 3).

#### 4.2.2. Correlations between the NO and the Number of Bacteria from Particular Families

The study of the correlation between the NO value and the recorded number of bacteria from the Gram + (cocci), Gram − (Cocci), and Gram + rods (mycobacteria) families in the control group was performed using Spearman’s rank correlation coefficient. The significance test showed no statistically significant correlations between the level of NO and the number of bacteria from individual families in the population of vaccinated people and unaffected (Table 4).

### 4.3. Comparison of Study and Control Group

#### 4.3.1. Comparison of NO Level in Exhaled Air

Statistical analysis of the results showed statistically significant differences in the distribution of NO in the populations considered: the vaccinated population and the infected, vaccinated and non-vaccinated people (*p*-value is less than 0.00001) (Table 5). Significantly higher NO values were present in the vaccinated and COVID-19-positive history population (sample rank average 35.5) compared to the vaccinated and with no COVID-19 history population (sample rank average 15.5). The arithmetic mean NO in the study group (41.8) is also higher than in the control group (21.7) (Figure 1).

#### 4.3.2. Comparison of Microbiome of Oral Cavity and COVID-19

The chi-square test of independence showed that COVID-19 infection does not have a statistically significant effect on the number of Gram + (cocci), Gram - (cocci), and Gram + rods bacteria (Table 6).

## 5. Discussion

A method for monitoring inflammation in the lungs–the assessment of NO concentration in the exhaled air is increasingly important in diagnosis and treatment monitoring [47,48,49,50]. An increase in NO concentration is associated with the exacerbation of inflammation and its decrease–with a positive therapeutic effect. Determining the level of NO in exhaled air is a method primarily recommended for monitoring the course of asthma [51,52], but it is also used in: allergic rhinitis, eosinophilic bronchitis, Churg-Strauss syndrome, atopic people, as well as in respiratory disorders during sleep [53,54]. It is also known that high concentrations of NO are found in the paranasal sinuses and the gastrointestinal tract [55,56]. The increase in NO synthesis was also demonstrated within the oral cavity tissues. However, the research material was saliva or pathologically changed mucosa, not exhaled air. AC. Batista et al., after taking a section of the mucous membrane covered with chronic periodontitis, showed that the level of NO was statistically significantly higher in patients with inflammation [57]. Oral diseases such as lichen planus (OLP) and recurrent aphthae (RAU) also cause an increase in NO concentration, but in saliva, which was confirmed by Masaru Ohashi et al. [58]. As with lichen planus or recurrent aphthae, this happens in Behcet’s syndrome, which was studied by Mukaddera Kocak et al. The authors presented a significantly higher level of NO in the collected material in patients suffering from Behcet’s syndrome [59].

Determining the level of NO in the exhaled air in pathological conditions of the oral cavity is a pioneering diagnostic method, and it is rarely used in dentistry. Literature reports on this subject are poor. The studies conducted so far have shown a significant impact of the presence of caries, periodontal inflammation, and poor oral hygiene on the level of NO concentration in exhaled air [60]. Such a relationship was also demonstrated among patients using complete denutres. The hygiene status of the dentures and the mucous membrane of the denture base significantly influenced the concentration of the inflammatory marker in the air. The most significant impact on the level of NO concentration in the exhaled air turned out to be teeth with active carious foci. In the case of patients with proper hygiene, the level of NO concentration was low [60]. Wyszyńska et al. described a case of meaningly elevated NO levels in her patient with peri-implantitis. After the elimination of inflammatory foci in the oral cavity, NO concentration in exhaled air decreased.

Based on the described case, the authors drew attention to the usefulness of the NO measurement device in the exhaled air, as well as the essence of the dental examination and possible elimination of odontogenic foci, which may influence the results of diagnostics of general diseases and treatment [61,62]. In the studies presented in this publication, the value of NO in particular groups of bacteria did not differ statistically significantly. The study of the correlation between the NO value and the recorded number of bacteria was also not significant. The results of the study concerned two groups of vaccinated patients with a positive and negative history of COVID-19. M.P. Hezel et al. described the relationship between the oral microbiome and NO. They drew attention to the ability of bacteria living in the oral cavity to produce NO in a situation of reduced oxygen supply. These bacteria, by reducing nitrates, lead to an increase in the amount of nitrogen compounds in saliva [63]. The role of the NO cycle in respiratory diseases was studied by Svetlana Soodaeva et al. They described the process of NO release by some species of bacteria by denitrification as a natural respiration process of anaerobic bacteria [64]. The influence of the oral microbiota on the so-called NO cycle is unfortunately not well understood yet, and the literature on NO and its correlation with the oral microflora is very poor. NO was also determined in terms of its distribution in the studied populations depending on the history of COVID-19.

Significantly higher NO values were obtained in the population of vaccinated and ill people compared to the control group. Measurements of NO in exhaled air were a topic of research by Paolo Cameli et al. The authors determined NO in patients with the so-called “post-COVID-19”. The values of exhaled NO in patients with a positive history of COVID-19 were significantly higher compared to the control group [65]. Similar research results on NO and COVID-19 were obtained by Bugra Kerget et al. CRP with severe symptoms of COVID-19 and significantly higher levels of NO in the examined patients [66]. The SARS-CoV-2 virus that causes COVID-19 is the subject of many scientific studies, but many of its negative effects are still unknown. Some researchers like Sonia Villapol et al. from the Houston Medical Institute, a center that has analyzed thousands of cases of recovering from COVID-19, explain that complications in patients infected with coronavirus without symptoms can also occur. The study coordinator explained that the observed problems often resulted from damage to the nervous system caused by SARS-CoV-2. In our control group (patients with no clinical symptoms of infection or unaware they had COVID-19), we didn’t observe an increase in the level of NO in exhaled air [67].

The impact of COVID-19 on the oral microbiome has not been the subject of many scientific studies. John P. Haran et al. studied the oral microbiome in patients with long-term symptoms of so-called long COVID-19. They showed a significantly higher number of pathobionts and an increase in pro-inflammatory bacteria in the examined patients. At the same time, they noticed the impairment of anti-inflammatory cell metabolism [68]. In the presented research, the dependence of the oral microbiome on COVID-19 disease was also checked. Based on the statistical analysis, it was shown that being ill with COVID-19 does not have a statistically significant effect on the number of bacteria from the Gram + (cocci), Gram − (cocci), and Gram + rods families that have been isolated in the examined group of patients. Research on COVID-19 still requires a deeper analysis and observation of the consequences of this disease, its impact on all units of our body and methods of prevention. Characteristics of the persistence times of the SARS-CoV-2 virus were described by Cervino G. et al. Results regarding temperature, humidity and surface material are useful for setting guidelines for the prevention of COVID-19 infection and valuable aim of further research [69].

## 6. Conclusions

A positive history of COVID-19 doesn’t influence the microbiome of the oral cavity. There are statistically significant differences in the distribution of NO in the considered populations. Vaccinated and post-COVID-19 patients had higher levels of NO in exhaled air than vaccinated and with negative history of COVID-19 patients (patients with no clinical symptoms of infection or unaware they had COVID-19), which suggests that the measurement of NO in exhaled air can be a complementary, non-invasive diagnostic and inflammation monitoring method.

## Figures and Tables

**Figure 1 jcm-12-05556-f001:**
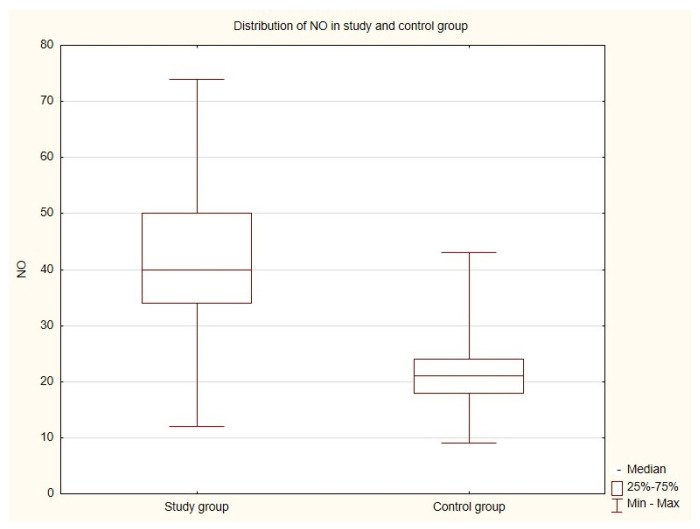
Statistically significant differences in the distribution of NO in the considered populations.

**Table 1 jcm-12-05556-t001:** Distribution of NO in relation to the number of bacteria (Kruskal-Wallis test—KW test)—the study group.

Bacteria Families	Study Group			Test	*p*-Value
	Number of Bacteria Group in Family	N *	Average Rank NO		
Gram + cocci	1	4	19.4	The Kruskal-Wallis Test	0.1074
	2	15	10.8		
	3	6	14.2		
Gram − cocci	0	1	20.0	The Kruskal-Wallis Test	0.3044
	1	16	11.4		
	2	8	15.3		
Gram + rods	0	10	12.0	The U Mann-Whitney Test	0.597795
	1	15	13.7		
Gram − rods	0	21	16.3	The U Mann- hitney Test	0.353599
	1 or 2	4	12.4		

* N—sample size.

**Table 2 jcm-12-05556-t002:** Correlations between NO and the number of bacteria from particular families (significance test for Spearman’s rank correlation coefficient)—study group.

Study Group: NO Level and Amount of Bacterias from Families:	N *	The Spearman’s Rank Correlation Coefficient	Significance Test for the Correlation Coefficient *p*-Value
Gram + cocci	25	−0.148	0.479097
Gram – Cocci	25	0.143	0.494002
Gram + rods	25	0.113	0.589528
Gram – rods	25	0.215	0.301805
Total study bacteria families	25	0.051	0.807809

* N—sample size.

**Table 3 jcm-12-05556-t003:** Distribution of NO in relation to the number of bacteria (Kruskal-Wallis test—KW test)—control group.

Bacteria Families	Study Group			Test	*p*-Value
	Number of Bacteria Group in Family	N *	Average Rank NO		
Gram + cocci	0 or 1	3	8.8	TheKruskal-Wallis test	0.4852
	2	18	14.0		
	3 or 4	4	11.6		
Gram − cocci	0	5	14.5	The Kruskal-Wallis test	0.5415
	1	14	11.6		
	2	6	15.1		
Gram + rods	0	12	16.0	The U Mann-Whitney test	0.052436
	1 or 2	13	10.2		

* N—sample size.

**Table 4 jcm-12-05556-t004:** Correlations between NO and the number of bacteria from particular families (significance test for Spearman’s rank correlation coefficient)—control group.

Control Group: NO Level and Amount of Bacteria Families:	N *	Spearman’s Rank Correlation Coefficient	Significance Test for the Correlation Coefficient *p*-Value
Gram + cocci	25	0.077	0.714696
Gram − Cocci	25	0.046	0.825976
Gram + rods	25	−0.348	0.088131
Total study bacterias families	25	−0.152	0.469167

* N—sample size.

**Table 5 jcm-12-05556-t005:** Comparison of NO values in the study group and the control group (Mann-Whitney U test—UMW test).

Group	N *	Arithmetic Mean NO	Standard Deviation NO	Average Rank NO	Test UMW *p*-Value
Study	25	41.8	14.3	35.5	<0.00001
Control	25	21.7	7.7	15.5	

* N—sample size.

**Table 6 jcm-12-05556-t006:** Influence of the disease on the number of bacteria (chi-square test of independence).

Bacteria Families	Number of Bacteria Group in Family	N *		Chi-Square Test of Independence *p*-Value
		Study	Control	
Gram + cocci	0 or 1	4	3	0.66512
	2	15	18	
	3 or 4	6	4	
Gram – cocci	0	1	5	0.21377
	1	16	14	
	2	8	6	
Gram + rods	0	10	12	0.56881
	1 or 2	15	13	

* N—sample size.

## Data Availability

Data supporting our results are available for request from the corresponding author.

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
