# Peer review of "The Impact of COVID-19 on the Oral Bacterial Flora in Patients Wearing Complete Dentures and on the Level of Exhaled Nitric Oxide as a Marker of Inflammation"

_jcm, 2023, doi:10.3390/jcm12175556_

Round 1
Reviewer 1 Report
Introduction: some note about nitric oxide, oral bacteria and a few sentences concerning COVID-19 in general.
Where is the connection between COVID-19 and nitric oxide? Why no references on possible connection are cited?
Where is the connection between total dentures and COVID-19? Why no references on possible connection are cited?
Where is the connection between total dentures and nitric oxide? Why no references on possible connection are cited?
Material and Methods
Why complete denture wearers are the target group and not the partial denture wearers? When complete denture wearers are a target group, why no dentated control group matching sex and age has been examined?
Had some denture wearers denture stomatitis? If not, it has to be an exclusion criterion. Denture stomatitis is not always clinically recognisable and can be responsible for exhaling of increased quantities of nitric oxide. Why study participants with systemic, respiratory, stomach and other inflammations have not been excluded? So, the non-consideration of denture stomatitis and other possible inflammations is a basic flaw completely compromising this study. Why no other exclusion criteria (e.g., medication) and no sex- and age-matched controls were applied?
Results: the oral bacteria are host symbionts and can widely vary, so an interconnection between the sorts of bacteria and mucosal inflammation cannot be deduced. So, the bacteriological examinations are redundant.
Discussion:
No relationship between COVID-19 and nitric oxide. A post-COVID-19 is not the same as COVID-19. The anamnesis is not a sufficient evidence for a post-COVID-19 not. There is no founded reason to measure nitric oxide in complete denture wearers with a post-COVID-19 anamnesis. The post-COVID-19 status is rather an object of infectious diseases and may include multiple pathologies. To search an relationship between denture wearers, nitric oxide and post-COVID-19 anamnesis.
The first two sentences of the abstract suffice: (i) Exhaled nitric oxide “seems” to be a “great” inflammation marker. This not only a clumsy English, it is a flawed one (ii). It increases in inflammation and decreases in healing process. The second sentence is redundant – so do all inflammatory markers.
Reviewer 2 Report
Dear Authors,I found this work impactful and fit well with in the scope of this journal. The manuscript needs some minor improvements; there are a few suggestions that authors may consider to improve it further:
The use of the English language is reasonable, however, there are a number of punctuation and grammatical errors; that should be corrected and rephrased using academic English for a better flow of text for reader.
Abstract: is precisely written, and the aim of the study is mentioned. Please include some more information about the results/finding to enhance the impact of this section.
The introduction; is detailed, compact, covering the background information and the rationale of the study effectively. However, the last paragraph is very details and suggested to condense that information.
Furthermore, In discussion, more studies in context should be included; as there is little support of literature from the previous studies.
I would suggest this paper:
Sars-cov-2 persistence: data summary up to q2 2020 DOI 10.3390/data5030081
· The conclusion section needs to be revised with a more clear and summarized outcome of the study.
I believe that your manuscript would have much more relevance after suggested improvements.
Round 2
Reviewer 1 Report
Anamnesis cannot replace the diagnosis – so wrong selection criteria were used. According literature, in most cases of abortive COVI-19 remains no pathology. Why not cite references supporting for hypothesis that abortive (or undiagnosed Covid-19) left behined some harms in humans?
